# OpenReview forum: "Compute as Teacher: Turning Inference Compute Into Reference-Free Supervision"
_ICLR.cc/2026/Conference — Submitted to ICLR 2026_

### Official Review · Reviewer_ySqQ · 2025-10-26

**Soundness:** 1
**Presentation:** 4
**Contribution:** 3
**Rating:** 4
**Confidence:** 4

**Summary:**

This paper proposes Compute as Teacher (CaT). CaT serves two functions: (1) it can be used as an inference-time algorithm to improve performance at test time and (2) it can be used to scale inference-time compute for RL while also enabling label-free training. This is enabled by using an anchor model `\pi_0` to aggregate policy "exploration" rollouts into a single response `s`. Crucially, the anchor model is prompted in such a way as to allow for disagreements with the majority consensus. This single response `s` may be returned directly during inference, or may be used as a reference for label-free training during RL. Importantly, CaT works in both the verifiable and the non-verifiable domains, with the latter enabled by self-written rubrics generated via conditioning on `s`.

To support their claims, the authors demonstrate that CaT yields improvements when used for RL training and as an inference-time algorithm. This is done by assessing three different models on two different datasets: MATH-500 (verifiable) and HealthBench (non-verifiable). The authors also demonstrate that (1) `s` may disagree with the consensus decision, and may even yield final answers that are absent in the original rollouts, (2) CaT + RL is superior to CaT + SFT and (3) CaT scales with the number of rollouts `G`, thereby enabling effective scaling of inference compute.

**Strengths:**

- This work aims to address three important issues faced by the community at the moment: (1) how do we improve model performance when no labels are provided? (2) how do we improve model performance using RL on non-verifiable tasks? and (3) how can we effectively scale compute during RL training? I appreciate the authors' attempt to tackle all of these problems simultaneously.
- While using aggregation as a technique for inference-time scaling is not new, few works explore using aggregation for reference-free training. Furthermore, the works that do (that I know of) rely on majority-voting mechanisms for aggregation, but this limits learning to "self-distillation" of majority consensus, which (at first glance) should not enable the trained model to solve new problems that the base model cannot already solve. CaT overcomes this by allowing the model to synthesise new aggregations that may differ from any of the original rollouts, which I believe is a novel contribution.
- The writing and overall presentation is very clear.

**Weaknesses:**

While the proposed method is promising, my main concern is that the experimental evidence provided is somewhat limited. The authors only evaluate on two datasets: MATH-500 (verifiable rewards) and HealthBench (non-verifiable rewards):
- MATH-500 is a relatively older and easier dataset that the base models (or at least, the instruct versions of those models) are able to attain very high scores on already: this is evidenced by the very small gap between "CaT"/"majority vote" and "single sample" for the strongest model (Qwen3-4B) in Figure 6, which suggests that consensus-based algorithms are hardly useful at all on this dataset to begin with. I would like to see further evidence of CaT's effectiveness on a larger number of and perhaps also more challenging collection of test sets (e.g. AIME, HMMT, LiveCodeBench etc.), where I believe your method should be more effective anyways. I would also like to see more RL baselines (e.g. RL with majority vote, RL with perplexity-based methods). I see related results in Table 6 but I assume these are for CaT at inference time rather than during RL - please let me know if I am mistaken.
- The authors train and evaluate CaT on the same dataset (page 6: "For MATH-500, we train and test on the same 500 questions..."). While I appreciate the fact that training is reference-free and so there is no risk of label memorisation, this still makes for an unfair compute-controlled comparison with the baselines: for a given test point `x_i`, you have already spent train-time compute "figuring out" (via exploratory rollouts) the correct answer to this point, which you then learn from via RL. As such, I believe the authors should report experimental results on a held-out test set in addition to the train set, or account for this in the baselines (by using relatively more test-time compute for those experiments).
- HealthBench is a domain specific dataset. While this is not an issue, I believe that the experimental evidence could be strengthened by including a more diverse set of instruction-following benchmarks (e.g. MT-Bench).
- The experimental results for HealthBench compare CaT with (1) RL + model-as-a-judge, where the model acts as a reference-based judge comparing `s` with each rollout and (2) RL + physician-written rubrics. The missing baseline here is RL with self-written rubrics that are independent of `s` (i.e. generate each rubric conditioned only on the question, as with existing rubric-related work), as this will tell us whether the effectiveness of CaT RL in the non-verifiable domain is due to the core CaT algorithm (generating `s` and using this as a pseudo-reference) or merely due to rubric-based training.

A second weakness is a lack of analysis demonstrating how CaT drives improvements:
- The authors claim that CaT improves upon existing baselines by allowing the model to learn from references that may be different to anything in the rollout set/in the minority (Page 2: "Parallel rollouts diversify partial competencies and different generations surface different sub-facts or solution steps...", "Conditioning the anchor on the set of rollouts
enables ensemble-like error correction within the model’s generative space..."). They provide a percentage (14%) of times this occurs (at inference-time?), but I would like to understand (1) during RL, when this does occur, how often is `s` actually correct? (2) how does this fraction change over the course of RL? (3) does the model ever solve previously "unsolvable" questions in `s` (i.e. does the aggregation mechanism enable the model to attain correct answers that could not be generated using normal rollouts under the policy?). I ask this because I would like to understand whether the primary mechanism driving learning is simply “sharpening” through training on the consensus answer (albeit with some noise ~14% of the time), or whether gains are instead driven by these diverse, minority answers.
- On the plateauing of performance: you argue that performance plateaus due to diversity collapse in the rollouts. This suggests that main driver of performance is indeed sharpening (whether the modal answers are reinforced). If this is the case, how does CaT meaningfully differ from consensus based RL-training and inference-time algorithms?
- How do CaT's outputs differ qualitatively from its policy rollouts? e.g. length, diversity of answers, reasoning strategies used etc. This may shed some light on how CaT qualitatively drives improvements.

**Questions:**

For me to increase my score to an acceptance, I would like to see my point about limited experimental evidence addressed through additional experiments. I may further increase my score given additional analysis on how exactly CaT drives improvement.

Some additional questions:
- How many replicates are used for the error bars in Figure 6?
- One of the downsides to CaT is that all `G` outputs must fit into context when generating `s`. This may cause problems for problems where the rollouts are longer. How might you circumvent this issue?

---

> ### Author Response · Authors · 2025-11-15
> **Response to Reviewer ySqQ**
>
> Many thanks for your feedback on our paper. We particularly appreciate the additional baselines you have suggested.
>
> > I would also like to see more RL baselines (e.g. RL with majority vote, RL with perplexity-based methods)
>
> We collected the results below on Llama 3.1 8B:
>
> | Method | HealthBench | MATH-500 |
> | :---- | :---- | :---- |
> | CaT-RL | **0.38** | **0.56** |
> | Majority Vote-RL | N/A | 0.53 |
> | min(PPL)-RL | 0.33 | 0.51 |
>
> Our analysis in Figure 6 correlates as expected with the RL results.
>
> > I would like to see further evidence of CaT's effectiveness on a larger number of \[...\] test sets
>
> We ran CaT across 3 seeds on HMMT and AIME’25 with Qwen 3 4B:
>
> | Method | HMMT | AIME’25 |
> | :---- | :---- | :---- |
> | Single sample | 6.7% | 10.0% |
> | Majority vote | 10.0% | 16.7% |
> | CaT | 13.3% | 20.0% |
>
> > The missing baseline here is RL with self-written rubrics that are independent of s
>
> We’ve run this below with Llama 3.1 8B:
>
> | Method | Accuracy |
> | :---- | :---- |
> | Rubric gen. conditioned on s | 0.38 +/- 0.01 |
> | Rubric gen. conditioned on q | 0.34 +/- 0.01 |
>
> Some improvement is possible with rubrics generated from q, though not as effective as conditioned on s. Both rubric-based training and the quality of the rubrics appear to contribute. Thank you for the suggestion.
>
> > They provide a percentage (14%) of times this occurs (at inference-time?), \[...\] when this does occur, how often is s actually correct?
>
> When CaT disagrees with consensus, it is correct on 69.6% of these occasions, so disagreement is useful for better supervision. Although we do not have statistics over the course of RL training, this is the supervision signal to the initial policy. Thank you as we want to show that CaT's disagreements are useful. We have revised the PDF to show this.
>
> > I ask this because I would like to understand whether the primary mechanism driving learning is simply “sharpening” through training on the consensus answer \[...\] or whether gains are instead driven by these diverse, minority answers.
>
> Our statistics show disagreement with majority vote is useful (see previous answer). The diverse minority answers provide better supervision over majority consensus. While rollout selection is inherently a sharpening mechanism by definition, in theory synthesis is better performing and could also go beyond sharpening by surfacing disagreeing perspectives and then targeting model thinking toward these disagreements to resolve them. In the minority disagreeing answers that are 69.6% correct, one could say they are no longer just “sharpening”.
>
> > How do CaT's outputs differ qualitatively from its policy rollouts?
>
> We have hinted at how CaT outputs differ in terms of resolving partial competencies in Appendix B. To investigate in more detail, we analysed CaT outputs and observed a few differences from rollouts.
>
> | Metric | Policy rollouts | CaT outputs | p-value (t-test) |
> | :---- | :---- | :---- | :---- |
> | Length (tokens) | 634 \+/- 11 | 639 \+/- 20 | 0.81 |
> | No. of equations | 32 \+/- 0.7 | 28 \+/- 0.7 | <0.01 |
> | Lines | 81 \+/- 2.2 | 53 \+/- 1.3 | <0.01 |
>
> CaT outputs have similar length to policy rollouts, despite achieving higher accuracy. This suggests CaT does not simply "think longer". Even among CaT outputs, correct answers are significantly shorter on average ($p<0.05$), further indicating that CaT benefits from reasoning already present in the rollouts, synthesizing relevant fragments rather than producing entirely new derivations. CaT outputs contain fewer equations and fewer lines than policy rollouts. Manual inspection indicates CaT often summarizes or enumerates possibilities that are already explored in the rollouts, rather than reconstructing steps. This behavior is consistent with CaT acting as a reconciling mechanism over the existing reasoning traces.
>
> We also examined stylistic markers of reasoning. Using keyword heuristics for step-by-step patterns (e.g., "first", "second"), we observe that CaT employs stepwise reasoning approximately 37% less often. Conversely, CaT outputs contain 9.3\% more verification cues (e.g., "let’s verify", "check").
>
> We have added Appendix C to the revised PDF with this analysis.
>
> > How many replicates are used for the error bars in Figure 6?
>
> We compute the standard error sample-wise over 500 samples for both test sets.
>
> > all G outputs must fit into context when generating s. This may cause problems for problems where the rollouts are longer. How might you circumvent this issue?
>
> Great observation. We see a few options: (a) sampling of a ‘best’ sub-group g from G via rule-based or LLM-based heuristics, (b) context compression where each policy rollout is summarised first, (c) random sampling of policy rollouts until the maximum context limit, or even (d) hierarchical aggregation.
>
> Thanks again for your detailed review. We have updated the revised PDF with all additional results. We are happy to discuss further if you have any more questions or concerns.

---

> ### Comment · Reviewer_ySqQ · 2025-11-17
> **Response to authors**
>
> Thank you for your detailed reply. I am glad to see that you have added additional baselines and analysis -- I will read through your responses and consider them in detail, and will update my score accordingly.
>
> From briefly scanning through your rebuttal, I notice that your results for HMMT and AIME have (1) scores that seem incorrectly low for Qwen-3 4B*, even when using non-thinking mode (see the Qwen technical report https://arxiv.org/pdf/2505.09388 for AIME results, and I also know from personal experience that the model should have higher HMMT scores) and (2) are probably benchmarked on too few seeds, as the datasets are very small.
>
> I would suggest you check your experimental setup here and also run on more (> 8) seeds.
>
> *The score discrepancies might be due different max response lengths. What response lengths are you using in your experiments?

---

> ### Author Response · Authors · 2025-11-17
> **Re: HMMT/AIME response lengths and running more seeds**
>
> Thank you for noting this discrepancy. You are correct that we use different max response lengths (1536 tokens) for the experiments we did on HMMT/AIME. It seems that the results collected for the Qwen technical report use a max response length of 38,912 tokens for AIME'25 in both thinking and non-thinking modes (page 13 of the technical report).
>
> We will get back to you with our results on >8 seeds. Given that Qwen 3 4B's context window is 128K and we use 8 rollouts in context with CaT, we are not able to match the technical report's response length (not all rollouts would fit). We could increase our response length up to 8K and run this experiment again for you. Would this be sufficient in your opinion?

---

> ### Comment · Reviewer_ySqQ · 2025-11-17
> **Response to authors**
>
> 1536 tokens seems a bit too short for these reasoning tasks (even for MATH-500?). I can think of two options here:
> (1) Rollout at 32k and use the summaries that Qwen-3 4B naturally generates (after < /think >) for CaT and discard the reasoning tokens (the stuff between < think > and < /think >).
> (2) Use Qwen-3-4B-Instruct and run at 8k or 16k. The model has a very long context window (200k I believe) so you might still be able to do CaT with that. The nice thing about Qwen-3-4B-Instruct is that its outputs are a bit less verbose than Qwen-3-4B and are typically <16k.

---

> ### Author Response · Authors · 2025-11-18
> **Results with 10 seeds**
>
> Thank you for offering these suggestions. We went with option (2) and ran Qwen3-4B-Instruct-2507 with response length 8k over 10 seeds. We also ran MATH-500 over 4 seeds with adjusted sampling settings (longer response length).
>
> | Method | HMMT | AIME’25 | MATH-500 |
> | :---- | :---- | :---- | :---- |
> | Single sample | 29.3 \+/- 1.56% | 42.7 \+/- 2.0% | 78.9 \+/- 0.4% |
> | Majority vote | 34.0 \+/- 1.39% | 55.3 \+/- 0.7% | 83.8 \+/- 0.4% |
> | Self-BoN | 36.7 \+/- 1.57% | 52.7 \+/- 1.9% | 83.2 \+/- 0.5% |
> | CaT | 36.7 \+/- 1.87% | 56.7 \+/- 2.1% | 87.3 \+/- 0.2% |
>
> CaT provides a modest but overall statistically significant (p<0.0001 against majority vote and p<0.001 against self-BoN) gain under meta-analysis (Appendix H) in this table, mainly driven by MATH-500. Please see the reply that follows for further evidence of CaT providing consistent gains across benchmarks and models.

---

> ### Author Response · Authors · 2025-11-20
> **Further experimental results**
>
> In addition to the above results, we would also like to highlights further results collected for reviewer mnEz that are also relevant to your concerns regarding limited experimental evidence.
>
> We collected results for Gemma 3 4B Instruct over 10 seeds (AIME'25) and 7 seeds (MATH-500). We also tried HMMT with 10 seeds, but Gemma could not achieve statistically significant results beyond 0%, so we exclude it:
>
> | Method | AIME’25 | MATH-500 |
> | :---- | :---- | :---- |
> | Single sample | 12.7 \+/- 0.8% | 71.7 \+/- 0.2% |
> | Majority vote | 11.3 \+/- 0.7% | 72.7 \+/- 0.3% |
> | Self-BoN | 14.0 \+/- 1.1% | 73.5 \+/- 0.2% |
> | CaT | 15.3 \+/- 0.9% | 77.4 \+/- 0.4% |
>
> Here are the disagreement rates for Qwen3-8B-Instruct-2507 for the results collected in the previous reply:
>
> |  | HMMT |  | AIME’25 |  | MATH-500 |  |
> | :---- | :---- | :---- | :---- | :---- | :---- | :---- |
> | Comparison | Disagreement rate | CaT correct when disagrees | Disagreement rate | CaT correct when disagrees | Disagreement rate | CaT correct when disagrees |
> | CaT vs Majority vote | 9.3 \+/- 1.6% | 65.0 \+/- 12.5% | 8.7 \+/- 0.9% | 57.5 \+/- 11.1% | 4.8 \+/- 0.2% | 86.3 \+/- 2.2% |
> | CaT vs Self-BoN | 6.7 \+/- 1.1% | 50.0 \+/- 14.4% | 6.7 \+/- 1.2% | 79.6 \+/- 8.2% | 6.5 \+/- 0.5% | 81.7 \+/- 3.7% |
>
> And for the Gemma 3 4B Instruct results:
>
> |  | AIME’25 |  | MATH-500 |  |
> | :---- | :---- | :---- | :---- | :---- |
> | Comparison | Disagreement rate | CaT correct when disagrees | Disagreement rate | CaT correct when disagrees |
> | CaT vs Majority vote | 4.7 \+/- 1.0% | 87.5 \+/- 12.5% | 7.4 \+/- 0.4% | 81.5 \+/- 2.7% |
> | CaT vs Self-BoN | 6.0 \+/- 0.8% | 64.8 \+/- 12.3% | 9.9 \+/- 0.7% | 70.2 \+/- 2.9% |
>
> CaT consistently outperforms majority vote and BoN baselines. It also consistently (in 4/5 cases) provides disagreements that are helpful (correctness >>50% when disagreeing).
>
> We also want to note that a significant contribution in our paper is the development of a method to enable RL fine-tuning from inference compute reference-free, even in non-verifiable domains. This can be used with any inference compute method (not only synthesis). Our experiments in verifiable domains serve as validation to demonstrate that synthesis is versatile, matching and outperforming baselines here as well as in the non-verifiable domain (our main focus). This means synthesis can be plug-and-play, working seamlessly between non-verifiable and verifiable domains. However, if there exists a better domain or task-specific inference strategy, this can be dropped in as a replacement while leveraging our RL reward mechanism (lines 498-500 in the paper).
>
> We hope these additional results have answered some of your questions and we thank you for your interest and continuing to engage in the discussion. Please let us know if you have any further concerns.

---

> ### Author Response · Authors · 2025-11-26
> **Update regarding experiments**
>
> Dear Reviewer,
>
> We wanted to follow up on the further experiments we posted in response to your reply last week. Based on your suggestions, we provided new experiments on Qwen and Gemma across AIME'25, HMMT, and MATH-500 with extended response length on up to 10 seeds as well as new statistics for these models and datasets on disagreement rates.
>
> Please let us know if this, as well as our earlier rebuttal response, has helped to address your concerns.
>
> Thank you, The Authors

---

### Official Review · Reviewer_mnEz · 2025-10-29

**Soundness:** 2
**Presentation:** 2
**Contribution:** 3
**Rating:** 4
**Confidence:** 3

**Summary:**

This work present Compute as Teacher (CaT) to utilize a model's exploration/stochasticity through parallel rollouts and a synthesis step to bootstrap a teacher signal from a frozen anchor. They compare their approach to different selection methods on both MATH-500 and HealthBench with three different models and show strong test-time and RL results.

**Strengths:**

The motivation for the work is strong, and a synthesis prompt offers a simple approach for bootstrapping from the frozen anchor model. Additionally, the distinct advantage of synthesis (may disagree with majority or find the correct solution even when all rollouts are incorrect) is compelling.

**Weaknesses:**

I mainly found issue with the presentation of the results. The graphical representation, while visually appealing, is difficult to parse and compare the results of different methods. In addition, the claims made do not seem to be fully supported by the corresponding results. Below are my main issues.

- Somewhere in the main text or appendix the actual values should be reported in a Table. Throughout the paper it is difficult to tell how CaT compares to the baseline approaches. For example, the format from Figure 8 that shows the raw accuracy scores instead of relative improvements would be more helpful for parsing the results.
- Result 1 does not seem to be true on MATH-500, as the error bars seem to overlap for all models. Even on HealthBench, the improvement seems to be marginal. Can the authors provide additional information on the significance of this gain?
- Similarly, the results in Figure 4 and 6 seem to be the same results - why are these not consolidated into one Figure/Table? CaT also underperforms or performs similarly to Self-BoN on 4/6 of the dataset-model pairs, but the authors claim that "CaT is superior to all baselines." Could the authors also provide additional information on the significance of this gain? I do not see the benefit based on the collected results of the synthesis prompt over just performing BoN.
- The SFT result is also unclear - it seems to not help at all compared to the initial model. Do the authors have motivation for why this is the case?
- Why is Figure 7 analyzed over only two models, and why are they two different models? Do results 5/6 hold across all models or just Gemma 3 4B and Qwen 3 4B, respectively? For example, it seems Majority would perform the same as CaT in Figure 7 (right) with Llama 3.1 8B.
- "Error bars are standard error." Forgive me if I am misunderstanding, but is this calculated across multiple rollouts (i.e. the standard error for the Initial model in Figure 4 is computed across N rollouts, and CaT is computed across 8N rollouts)? I could not find mention of what N is.

**Questions:**

Most of my questions are listed in the Weaknesses section.
- Why do the authors utilize the word 'combine' in the freeform prompt and 'synthesize' in the reasoning prompt? Also I am curious how robust the final performance is to the synthesis prompt.
- The authors claim "disagreement with majority on 14% of questions, and disagreement with all rollouts on almost 1%." Could the authors expand on this result? For example, how frequently does CaT disagree with BoN? Can these disagreements be broken down into any categories to better understand the benefits of CaT?

---

> ### Author Response · Authors · 2025-11-15
> **Response to Reviewer mnEz**
>
> Thank you for your review of our paper. We particularly appreciate your input regarding statistical significance.
>
> > Somewhere in the main text or appendix the actual values should be reported in a Table.
>
> We have now added these tables to Appendix A. In transcribing, we discovered a mistake in Figure 4 for Qwen 3 4B on MATH-500. The CaT and CaT-RL results are reversed. CaT-RL outperforms CaT\.
>
> > Result 1 does not seem to be true on MATH-500, as the error bars seem to overlap for all models.
>
> Our main claim is that CaT and CaT-RL both improve significantly over the initial policy. Even if CaT-RL performs only similarly to CaT, this means that at test-time, a CaT-RL trained model produces responses as good as CaT, but with approximately G+1=8+1=9x less compute.
>
> Nevertheless, for MATH-500, error bars touch only marginally and CaT-RL is consistently better. Across models and datasets, CaT-RL beats CaT significantly (p=0.021) under meta-analysis on standardized effect sizes. Please see Appendix H in the revised PDF for further details.
>
> > the results in Figure 4 and 6 seem to be the same results
>
> Figure 4 compares the RL method against initial and CaT. Figure 6 compares inference-time methods as potential supervision. The figures are separated to express different findings.
>
> >  [...] performs similarly to Self-BoN [...] Could the authors also provide additional information on the significance of this gain?
>
> Please refer to our new tables in Appendix A. CaT outperforms self-BoN on 5/6 cases. Pooling z-scores for meta-analysis with inverse variance weighting, CaT significantly outperforms self-BoN (p=0.017).
>
> > The SFT result is also unclear - it seems to not help at all compared to the initial model.
>
> Our results are consistent with [A, Table 1] where expert answer SFT does not improve results. It is therefore not surprising to us that SFT on synthesised answers also does not help. A potential reason is that RL often leads to superior generalisation and sample-efficiency over SFT \[B\]. Given the small size of the HealthBench training set, this could be why SFT is unhelpful in comparison. Further, RL on rubrics provides a softer target that does not penalise answers for using different wording / syntax while expressing the correct point.
>
> > Why is Figure 7 analyzed over only two models, and why are they two different models?
>
> It was an arbitrary choice to analyse over different models. We agree that it would be more sensible to stick to a single model or be thorough and demonstrate over all. We re-ran the scaling for MATH-500 with Qwen 3 4B as a quick check for consistency:
>
> | Rollouts | Accuracy |
> | :---- | :---- |
> | 1 | 0.80 |
> | 2 | 0.82 |
> | 4 | 0.84 |
> | 8 | 0.85 |
>
> The results indicate similar scaling, with a stronger plateau. For Llama 3.1 8B, you are right that it would perform the same as the majority vote. We have revised the PDF in orange to include this observation and that weaker/older models may not be as effective at reconciling/synthesising rollouts.
>
> > is this calculated across multiple rollouts
>
> Yes, N is the number of samples in the test set. CaT is computed across N rollouts rather than 8N however as the group of 8’s synthesised output is just 1 CaT output. For single sample, we compute the average over the first rollout and so the error is computed across N rollouts.
>
> > Why do the authors utilize the word 'combine' in the freeform prompt and 'synthesize' in the reasoning prompt?
>
> This was an arbitrary choice and not investigated. The prompt is relatively robust to phrasing but worth investigating further in future.
>
> > The authors claim "disagreement with majority on 14% of questions, and disagreement with all rollouts on almost 1%." Could the authors expand on this result? \[...\] how frequently does CaT disagree with BoN? Can these disagreements be broken down into any categories to better understand the benefits of CaT?
>
> At inference time, when CaT disagrees with majority vote, it is correct on 69.6% of these occasions. Against BoN, CaT disagrees less often (4.8% of the time), but when it disagrees, it is correct 79.2% of times. CaT disagrees with majority voting and BoN in a way that leads to a better supervision signal. In Appendix B, we qualitatively show why CaT is beneficial from total disagreement through a worked example where it corrects errors in the policy rollouts. Thank you for highlighting this. We have updated the revised PDF with these statistics.
>
> Thanks again for your review. We are very open to discussing any further questions or concerns you may have.
>
> \[A\] Gunjal, A., Wang, A., Lau, E., Nath, V., He, Y., Liu, B. and Hendryx, S., 2025\. Rubrics as rewards: Reinforcement learning beyond verifiable domains. arXiv preprint arXiv:2507.17746.
>
> \[B\] Chu, T., Zhai, Y., Yang, J., Tong, S., Xie, S., Schuurmans, D., Le, Q.V., Levine, S. and Ma, Y., 2025\. Sft memorizes, rl generalizes: A comparative study of foundation model post-training. arXiv preprint arXiv:2501.17161.

---

> > ### Comment · Reviewer_mnEz · 2025-11-18
> >
> > Thanks for the extra details. To summarize my thoughts, I do like the story and motivation behind this work, but I am unconvinced that synthesis is doing anything meaningful compared to simpler aggregation baselines (most notably BoN). Here are a few more questions/concerns to hopefully get at this further:
> > 1. Sample-wise standard error is unconvincing. I am less familiar with related work in this space than the other reviewers but I have not seen this done before. From my understanding, running over random seeds is possible here although computationally expensive. Could the authors please motivate this choice? Is there an assumption being made that each question in MATH-500 is independent, and why is that reasonable?
> > 2. In that vein, the authors did run additional results in their response to Reviewer ySqQ with multiple seeds. Could the authors also provide the standard deviation across the ten seeds on these datasets? And also provide the results for BoN on these datasets?
> > 3. From the intuitive example, I can understand how CaT *may* disagree with other aggregation schemes. However, with Qwen 3 4B on MATH-500, it only disagrees with BoN on 24/500 samples. Of these 24 samples, it is correct 19 times and incorrect 5 times (where the sample-wise standard error is $\pm10$?). Importantly, this also seems to be the model/dataset combination where CaT performs the best compared to BoN, so I am unsure if this supports the claim that CaT "surfaces disagreeing perspectives and targets the model toward thinking beyond simple plurality." Additionally, analysis of number of equations or lines is not particularly convincing for proving this claim. Could the authors collect Result 7 across models/datasets?

---

> ### Author Response · Authors · 2025-11-19
> **Response to additional comments**
>
> **Response to point (1)**
>
> We understand your concerns about this. You are right that for RL (and beyond), it is very computationally expensive to run over random seeds, so this is typically not done. For example, this recent NeurIPS 2025 (oral) paper \[A\] does not provide indication of error as far as we can tell. Other prominent examples include the DeepSeek R1 report \[B\], Dr. GRPO \[C\], and Absolute Zero \[D\]. To provide *some* indication of error without the additional compute burden, we provide the sample standard error. Although not perfect, we assumed independence of test questions is somewhat reasonable for diverse math problems.
>
> **Response to point (2)**
>
> Here are the results you asked for with mean and standard error over the 10 seeds. We also ran MATH-500 over 4 seeds with adjusted sampling settings (longer response length).
>
> | Method | HMMT | AIME’25 | MATH-500 |
> | :---- | :---- | :---- | :---- |
> | Single sample | 29.3 \+/- 1.56% | 42.7 \+/- 2.0% | 78.9 \+/- 0.4% |
> | Majority vote | 34.0 \+/- 1.39% | 55.3 \+/- 0.7% | 83.8 \+/- 0.4% |
> | Self-BoN | 36.7 \+/- 1.57% | 52.7 \+/- 1.9% | 83.2 \+/- 0.5% |
> | CaT | 36.7 \+/- 1.87% | 56.7 \+/- 2.1% | 87.3 \+/- 0.2% |
>
> We acknowledge that on harder datasets, the variance is high due to the small problem set. However, the consistent positive signal across datasets suggests CaT provides a stable improvement.
>
> We would like to note a few important things at this point. Firstly, though gains are modest, inverse-variance weighted analysis indicates CaT outperforms majority vote (p\<0.0001) and self-BoN (p\<0.001) in this table, mainly driven by the improvement in MATH-500 which has the smallest error bars. In our earlier response, we also showed that CaT’s gains were overall statistically significant with Qwen, Gemma, and Llama across HealthBench and MATH-500. Particularly in HealthBench, we see larger and consistent improvements over self-BoN.
>
> Crucially, a significant contribution in our paper is the development of a method to enable RL fine-tuning from inference compute reference-free, even in non-verifiable domains. This can be used with any inference compute method (not only synthesis). Our experiments in verifiable domains serve as validation to demonstrate that synthesis is versatile, matching and outperforming baselines here as well as in the non-verifiable domain (our main focus). This means synthesis can be plug-and-play, working seamlessly between non-verifiable and verifiable domains. However, if there exists a better domain or task-specific inference strategy, this can be dropped in as a replacement while leveraging our RL reward mechanism (lines 498-500 in the paper).
>
> **Response to point (3)**
>
> We computed result 7 again over the results we collected for point 2\.
>
> |  | HMMT |  | AIME’25 |  | MATH-500 |  |
> | :---- | :---- | :---- | :---- | :---- | :---- | :---- |
> | Comparison | Disagreement rate | CaT correct when disagrees | Disagreement rate | CaT correct when disagrees | Disagreement rate | CaT correct when disagrees |
> | CaT vs Majority vote | 9.3 \+/- 1.6% | 65.0 \+/- 12.5% | 8.7 \+/- 0.9% | 57.5 \+/- 11.1% | 4.8 \+/- 0.2% | 86.3 \+/- 2.2% |
> | CaT vs Self-BoN | 6.7 \+/- 1.1% | 50.0 \+/- 14.4% | 6.7 \+/- 1.2% | 79.6 \+/- 8.2% | 6.5 \+/- 0.5% | 81.7 \+/- 3.7% |
>
> Despite the large error bars in HMMT/AIME as the number of samples where disagreements happen is a subset of an already small dataset, on AIME’25 CaT seems to be correct much more often than Self-BoN when it disagrees (far above chance). Similarly, as the error bars are much smaller for MATH-500, these results are also significant. Overall inverse variance-weighted analysis, mainly driven by MATH-500’s small error bars, indicates CaT disagrees in a useful way beyond chance (p\<0.001) against both majority vote and self-BoN.
>
> Thank you for your comments and for your engagement in this discussion. We will provide further results on additional models for result 7 as soon as we can. We hope this has helped to clarify your questions.
>
> \[A\] Yue, Y., Chen, Z., Lu, R., Zhao, A., Wang, Z., Song, S. and Huang, G., 2025\. Does reinforcement learning really incentivize reasoning capacity in LLMs beyond the base model?. NeurIPS 2025\.
>
> \[B\] Guo, D., Yang, D., Zhang, H., Song, J., Zhang, R., Xu, R., Zhu, Q., Ma, S., Wang, P., Bi, X. and Zhang, X., 2025\. Deepseek-R1: Incentivizing reasoning capability in LLMs via reinforcement learning. arXiv preprint arXiv:2501.12948.
>
> \[C\] Liu, Z., Chen, C., Li, W., Qi, P., Pang, T., Du, C., Lee, W.S. and Lin, M., 2025\. Understanding R1-Zero-like training: A critical perspective. COLM 2025\.
>
> \[D\] Zhao, A., Wu, Y., Yue, Y., Wu, T., Xu, Q., Lin, M., Wang, S., Wu, Q., Zheng, Z. and Huang, G., 2025\. Absolute Zero: Reinforced self-play reasoning with zero data. arXiv preprint arXiv:2505.03335.

---

> ### Author Response · Authors · 2025-11-20
> **Response to additional comments part 2**
>
> Thank you for your patience. In addition to our previous follow up, we have now collected result 7 with Gemma 3 4B instruct across 10 seeds for AIME/HMMT and 7 seeds for MATH-500. We note that Gemma does not perform much beyond 0% on HMMT (max. 1.3% with results largely within margin of error) so we exclude it in the following:
>
> | Method | AIME’25 | MATH-500 |
> | :---- | :---- | :---- |
> | Single sample | 12.7 \+/- 0.8% | 71.7 \+/- 0.2% |
> | Majority vote | 11.3 \+/- 0.7% | 72.7 \+/- 0.3% |
> | Self-BoN | 14.0 \+/- 1.1% | 73.5 \+/- 0.2% |
> | CaT | 15.3 \+/- 0.9% | 77.4 \+/- 0.4% |
>
> |  | AIME’25 |  | MATH-500 |  |
> | :---- | :---- | :---- | :---- | :---- |
> | Comparison | Disagreement rate | CaT correct when disagrees | Disagreement rate | CaT correct when disagrees |
> | CaT vs Majority vote | 4.7 \+/- 1.0% | 87.5 \+/- 12.5% | 7.4 \+/- 0.4% | 81.5 \+/- 2.7% |
> | CaT vs Self-BoN | 6.0 \+/- 0.8% | 64.8 \+/- 12.3% | 9.9 \+/- 0.7% | 70.2 \+/- 2.9% |
>
> Here too, CaT consistently outperforms majority vote and BoN baselines. It also consistently provides disagreements that are helpful (correctness >>50% when disagreeing).
>
> Thank you for your informative follow up questions. Please continue to let us know if you have any further concerns.

---

> ### Author Response · Authors · 2025-11-26
> **Update regarding experiments**
>
> Dear Reviewer,
>
> We wanted to follow up on the further experiments we posted in response to your reply last week. Based on your suggestions, we provided new experiments on Qwen and Gemma across AIME'25, HMMT, and MATH-500 on up to 10 seeds as well as new statistics for these models and datasets for result 7.
>
> Please let us know if this has helped to address your concerns and if so, we would be grateful if you could update your score to reflect this.
>
> Thank you, The Authors

---

> > ### Comment · Reviewer_mnEz · 2025-11-26
> >
> > Thanks! A few more quick questions.
> >
> > > To provide some indication of error without the additional compute burden, we provide the sample standard error. Although not perfect, we assumed independence of test questions is somewhat reasonable for diverse math problems.
> >
> > I do not agree that this is better, as it gives a false impression of the standard deviation under faulty assumptions and I think the paper would be better off not reporting sample standard error at all (but perhaps the other reviewers and AC can weigh in).
> >
> > >  adjusted sampling settings (longer response length).
> >
> > Could the authors explain this choice? Also, the reported Qwen 3 4B CaT result on MATH-500 is 81% in Table 3 but 87.3% +- 0.2% in the rebuttal response. Could the authors explain this discrepancy? Apologies if I am misreading something somewhere.
> >
> > Overall, I'm quite borderline on this paper. The story is nice and I can appreciate the utility of "RL fine-tuning from inference compute reference-free". However, from both the initial results from the paper and the additional experiments collected during the rebuttal phase, the framework does not seem to offer a consistent performance improvement over simpler baselines.

---

> ### Author Response · Authors · 2025-11-26
> **Additional response to Reviewer mnEz**
>
> Thank you for your follow up and additional questions.
>
> **In response to point (1)**, we are happy to remove sample-wise error bars if these assumptions are faulty. We note that it is common to estimate errors sample-wise over the dataset in similar work, e.g., via bootstrap sampling.
>
> **In response to point (2)**, we used longer response lengths for these follow-up experiments as this was requested by Reviewer ySqQ to match their expected sampling configuration for the new datasets (AIME'25/HMMT) and for MATH-500. Since you asked a similar follow up question, we provided the same results to you. This allowed us to provide results for both you and Reviewer ySqQ without running these computationally expensive multi-seed experiments again. The results are naturally a little better because of the extended response/reasoning lengths.
>
> > However, from both the initial results from the paper and the additional experiments collected during the rebuttal phase, the framework does not seem to offer a consistent performance improvement over simpler baselines.
>
> **Regarding performance**, we respectfully disagree. In the new results provided, CaT beats majority vote in 3/3 cases for Qwen and 2/2 for Gemma. Similarly, CaT beats BoN in 2/3 cases for Qwen and 2/2 for Gemma. 9/10 wins for CaT against the simple baselines is significant under a sign test (p=0.01). Moreover, among disagreement rates, CaT is more right than wrong to disagree with majority vote in 3/3 cases (Qwen) and 2/2 cases (Gemma). CaT is as good or more right than wrong to disagree with BoN in 3/3 cases (Qwen) and 2/2 cases (Gemma). In our view, these results indicate consistent improvement over simpler baselines.
>
> Beyond the CaT results, CaT-RL also performs significantly beyond the initial policy. If the reviewer is hesitant to agree that CaT-RL outperforms CaT and only matches it, then this result still shows that models trained with CaT-RL can produce outputs of the same quality as CaT with 9x less compute (no need to sample group rollouts at test time). Furthermore, CaT-RL is a drop-in approach. Self-generated rubric verification may be used for RL training with estimated references from the simpler inference selection methods if desired (not only CaT).
>
> Thank you again for your followups and continued engagement. Have our answers above clarified your questions?

---

> > ### Comment · Reviewer_mnEz · 2025-11-26
> >
> > Thanks for the response! And the result in Table 6 is with the longer or shorter response lengths? It seems that the rank ordering is different in Table 6 vs the rebuttal result (BoN does not benefit from the extended response/reasoning lengths?). This is my main confusion as the benefit of CaT over BoN/Majority Vote is clearest on this dataset.
> >
> > > models trained with CaT-RL can produce outputs of the same quality as CaT with 9x less compute (no need to sample group rollouts at test time)
> >
> > I agree that this is an interesting result, thank you for reiterating it.

---

> ### Author Response · Authors · 2025-11-26
>
> Thank you. Table 6 is the older result with the shorter response lengths. Indeed, while Self-BoN improves, it seems like it has not benefitted as much as majority vote from the extended response lengths in the newer rebuttal results. Since the newer results are multi-seed, these provide a more reliable estimate of rank order.
>
> We conjecture that it may be more difficult to select the best rollout with self-BoN when responses are longer as there is more context to reason about, hence why it may not benefit as much as majority vote, but still improves because each rollout is more likely to be correct. You have highlighted an interesting point. It would be valuable future work to further test how these methods scale with response length. Thank you.
>
> Is there anything further we can clarify?

---

> ### Author Response · Authors · 2025-11-27
> **PDF update for clarity and new results**
>
> To ensure the distinction between the short-response (Table 6) and long-response (rebuttal) settings is clear for future readers, we have updated the appendix to explicitly contrast these results in Table 6 and Table 7 (now containing the new and complete rebuttal results).
>
> We have now also updated Table 2 in the revised PDF to show our disagreement statistics, now run over multiple seeds as you suggested.
>
> We hope this final addition further clarifies our method and resolves the confusion. Regarding your remaining source of uncertainty in your assessment, we would be grateful if you could consider adjusting your rating if you feel this is now resolved.

---

### Official Review · Reviewer_wyte · 2025-10-31

**Soundness:** 3
**Presentation:** 2
**Contribution:** 2
**Rating:** 4
**Confidence:** 3

**Summary:**

This paper introduces Compute as Teacher (CaT), a method that converts inference-time compute into reference-free supervision. For each prompt, a current policy model generates multiple parallel rollouts, which are then synthesized by a frozen anchor model (the initial policy) into a single estimated reference answer. This synthesized reference serves as the supervision signal. The authors designed two regimes:
(a) **Verifiable tasks** (e.g., mathematics) — rewards are computed via programmatic equivalence checks; (b) **Non-verifiable tasks** (e.g., open-domain dialogue, health advice) — the model generates self-proposed rubrics, binary evaluation criteria scored by an independent LLM judge.

The paper tests both inference-time synthesis (CaT) and reinforcement learning with synthesized supervision (CaT-RL). Experiments across Gemma 3 4B, Qwen 3 4B, and Llama 3.1 8B on MATH-500 and HealthBench show substantial improvements. CaT outperforms selection and self-consistency baselines, demonstrating that reinforcement learning with synthetic supervision can improve beyond the initial synthesized teacher.

**Strengths:**

Reviewer found the following strengths of the paper:

**- Novelty and Conceptual Contribution**:
CaT proposes a new way to transform inference computation (different rollouts generated by the model) into self-supervision without human references, bridging selection-based self-consistency and self-play reinforcement learning. The idea of synthesizing multiple rollouts via a frozen anchor, rather than selecting one, is conceptually new.

**- Experimental Breadth and Depth**:

The authors test both inference-time and training-time variants across multiple models and tasks, providing thorough empirical validation.
Ablation and mechanism analyses (e.g., scaling with rollout number, disagreement analysis) are comprehensive and align with claims.

**- Baseline Coverage:**
The comparisons span single-sample, best-of-N, perplexity, mutual predictability, majority voting, and SFT methods, etc, showing consistent and notable gains for CaT.

**Weaknesses:**

Though the paper is suffering by the following issues:

**a. Heavy Reliance on Prior Terminology and Concepts**: The paper is **difficult to follow because it assumes extensive prior knowledge** of recent LLM post-training methods, such as rubrics, self-editing, and test-time-compute. These concepts are used without sufficient background or intuitive explanation, especially in the **Introduction**. As a result, readers unfamiliar with recent self-training or LLM-as-a-judge literature may struggle to understand how CaT differs from prior work or why it is a conceptual advance. Such presentation styles thus weaken the clarity of the paper’s core contribution.

**b. Lack of Clear Intuition Behind the “Synthesis"**: While the synthesis of multiple rollouts plays as a core idea of the paper, **why** we should do it to yield a better reference is not fully justified. The authors instead mostly describe how synthesis is done but provide little intuitive reasoning or fundamental reasons to expect better results than selection or ensembling. Without a clear and convincing insight, **the synthesis idea feels somewhat heuristic or ad-hoc**.

**c. Reward Reliability in Non-Verifiable Tasks and Dependence on Anchor Model Quality**: Rubric-based rewards are promising but rely **heavily on the accuracy and stability of the LLM judge**. The paper indeed does not analyze the sensitivity to judge model choice or rubric quality over training iterations. Furthermore, the effectiveness of CaT critically depends on the frozen anchor model’s capacity to perform meaningful synthesis over multiple rollouts. If the anchor lacks sufficient reasoning or summarization ability, the synthesized output may simply mirror the same errors or contradictions rather than correcting them. In such cases, the supervision signal could become noisy or even misleading. The paper has not studied how anchor strength affects performance, leaving it unclear to the
Review whether gains come from the CaT mechanism itself or from the anchor’s inherent capability.

**d. ** Compute and Efficiency Concerns**: Although the authors claim low overhead, the synthesis and rubric evaluation steps may introduce significant inference costs. Thus, quantitative analysis of the compute-to-performance trade-off is important to strengthen the argument.

**Questions:**

Given the above waekness (except the terminology presentation), here are the questions that the Reviewer be confused:

- What is the underlying intuition for why synthesizing multiple rollouts should produce a better answer than selecting the best one (e.g., via majority vote or confidence scoring)?

- How sensitive is CaT’s performance to the quality or scale of the frozen anchor model? If a weaker model is used as the anchor, does the synthesized reference still improve over rollouts, or does performance degrade?

- For non-verifiable domains, how stable are rubric-based rewards across different judge models (e.g., GPT-4o vs. smaller open models)?
Do rubric criteria drift or degrade as the policy improves over training?

- How does the additional inference and judging cost of CaT compare quantitatively to standard best-of-N or self-consistency methods?

---

> ### Author Response · Authors · 2025-11-15
> **Response to Reviewer wyte**
>
> Thank you for taking the time to write a detailed review. We are happy that you noted the novelty of our contribution, experimental breadth and depth, and baseline coverage. Below, we have provided responses to your questions and concerns:
>
> > The paper is difficult to follow because it assumes extensive prior knowledge
>
> We have taken the time to add further extensive definitions and additional examples to the introduction. Please see the changes in orange in the revised PDF.
>
> >  Lack of Clear Intuition Behind the “Synthesis"
>
> In lines 16-18, we explain that “a frozen anchor (the initial policy) reconciles omissions and contradictions to estimate a reference” through synthesis. In the introduction we also have a “Why it works (intuition)” paragraph on lines 94-101. Is there anything further you would like us to clarify regarding the intuition? This would greatly help us communicate our ideas more clearly.
>
> > The paper indeed does not analyze the sensitivity to judge model choice or rubric quality over training iterations.
>
> Great question. We ran a preliminary RL run with Llama 3.1 8B on the sensitivity of the rubric verifier/judge model. We used Llama 3 70B for evaluation as we are limited on compute budget.
>
> | Rubric verifier / Judge model | HealthBench accuracy |
> | :---- | :---- |
> | Llama 3.1 8B | 0.48 |
> | Llama 3.3 70B | 0.54 |
>
> This seems to indicate that stronger rubric verifiers/judges lead to better outcomes on non-verifiable rubric RL. Thank you.
>
> > it unclear to the Review whether gains come from the CaT mechanism itself or from the anchor’s inherent capability.
>
> Please refer to Appendix A in the revised PDF for additional clarification. We show that CaT is significantly better than the initial policy in all cases and thus the CaT mechanism is the driver of performance and not the anchor’s inherent capability. Moreover, regarding the RL, pooling z-scores for CaT-RL vs CaT results (weighting inversely by variance) \= 2.31 (p=0.021) which is also significant. These results show CaT-RL beats CaT. Thus, against the initial policy, CaT drives gains, and for CaT-RL, our RL method drives gains against CaT.
>
> > may introduce significant inference costs
>
> This is a valid concern, but we do not think it applies for our approach. For important queries like health, trading off speed for quality seems reasonable if using CaT at inference. If using a model trained via CaT-RL, it can be applied at inference with the same inference cost as the initial model (there is no significant difference in output length in terms of tokens), and thus, there is no inference tradeoff when using a model trained with CaT-RL.
>
> > How sensitive is CaT’s performance to the quality or scale of the frozen anchor model? If a weaker model is used as the anchor
>
> In Figure 7, Llama 3.1 8B, which is an older/weaker model compared to Gemma 3 4B and Qwen 3 4B, does not leverage rollout diversity to beat majority voting. This implies that weaker models may not be as capable in terms of synthesising outputs to improve answers relative to the stronger Gemma and Qwen models. We have revised the paper to mention this insight in orange. Thank you.
>
> > For non-verifiable domains, how stable are rubric-based rewards across different judge models \[...\] Do rubric criteria drift or degrade as the policy improves over training?
>
> We did observe that performance does degrade eventually over training. This is why we used a train-validation-test split for HealthBench where we used early stopping at peak validation accuracy and measured generalisation on the test set at this point.
>
> > additional inference and judging cost of CaT compare quantitatively to standard best-of-N or self-consistency methods?
>
> The complexity of CaT is similar to that of best-of-N or majority voting. In terms of input tokens, these scale with O(G) where G is the size of the group of trajectories. The number of output tokens in CaT does not scale with G (as it outputs a single answer/trace) and is amortised as G increases. Thus, it scales just as well in terms of cost as these methods.
>
> Thank you again for your review. We would be happy to respond to any further questions or concerns you may have.

---

> ### Author Response · Authors · 2025-11-25
> **Update regarding experiments**
>
> Dear Reviewer,
>
> We wanted to follow up on the rebuttal we posted 10 days ago. Based on your suggestions, we provided new experimental analysis on the effect of the judge model, a pointer to the intuition behind synthesis, tabular results and statistical tests to show that CaT does indeed outperform the initial policy and that CaT-RL outperforms CaT, as well as answers to your other questions.
>
> Could you please let us know if this addresses your concerns? If so, we would be grateful if you could update your score to reflect this.
>
> Thank you, The Authors

---

> > ### Comment · Reviewer_wyte · 2025-11-25
> > **Response to Authors**
> >
> > Thank you to the authors for addressing my concerns. The new version seems to be an improvement, especially in the introduction. While I'm still not really satisfied with the novelty or convinced by the intuition of the paper, I will increase the score given, as most of my concerns are discussed and addressed to some extent.

---

> > > ### Author Response · Authors · 2025-11-26
> > > **Thank you Reviewer wyte**
> > >
> > > We are glad to hear that we have addressed most of your concerns. We will work to address the remaining points. Thank you for your support of our paper.

---

### Official Review · Reviewer_3rfK · 2025-11-04

**Soundness:** 4
**Presentation:** 4
**Contribution:** 3
**Rating:** 6
**Confidence:** 4

**Summary:**

The paper asks whether compute can serve as a teacher in environments where it is either hard to get post-training data or there is no ground truth. The authors demonstrate that this can be done using CaT and CaT-RL, which generate many rollouts and put these rollouts in context without the original question and ask the original model to synthesize the answers, upon which it is graded using either a self-generated rubric or verifiable rewards. In both cases, CaT does impressively in datasets and beats various baselines in a comprehensive set of analyses.

**Strengths:**

Paper is concise and extremely well written. Broader research question is well motivated within the literature and the past work sections help clear up how the paper uniquely contributes compared to other works. Improvements over baselines are high and results and analyses tell a cohesive and convincing story about how compute can serve as a reward signal.

**Weaknesses:**

My biggest complaint is that CaT, as constructed, seems a bit arbitrary in its various design decisions. I get that this isn't a problem if the overarching paper story is more towards whether compute CAN serve as rewards---as it is simply a demonstration that it can, but if the framing is more towards using CaT as a general framework then I have some issues with it.

1. There isn't direct evidence provided by the authors that not including the question in the synthesis prompt actually does anything (or maybe I missed it?). The authors present an intuitive argument but never provide an ablation comparison on CaT but with including the question text in context along with the rollouts. Thus, it feels like an arbitrary decision that isn't justified if you want to propose CaT as a general framework.

2. In the self-generated rubric, there are also some seemingly-arbitrary design decisions. For instance, limiting to yes/no questions might be less expressive than likert scale questions or similar finer-grained ratings. Why not try those instead if model capabilities permit this?

3. Another detail is that one could imagine doing the rollouts with different prompt rephrasings or conditioning on other differing information in order to encourage exploration, which the authors point out as an issue.

4. Lastly, simply aggregating by putting everything into context may not be the best at larger rollout numbers --- maybe asking the model to filter for diverse responses will yield better combinations?

Overall, these seem like design decisions that don't fully justify the authors' central focus of CaT as a general framework, but do answer the research question of whether you can use compute as training in general. Therefore I would personally favor that framing.

**Questions:**

1. "Since CaT resolves disagreements and omissions to produce better estimated references, it can no longer improve over the individual rollouts if they tend to agree too much."
Do the authors have any ideas on how to potentially alleviate this? Otherwise CaT seems like a one-time boost to any model family rather than a stable method to continuously improve models (which of course is still valuable, but the latter is much more extremely so).

2. I don't really buy that CaT can meaningfully scale with rollouts beyond a certain extent; do the authors believe there will be settings where there are meaningful increases after a reasonably large magnitude (e.g., 20)? Maybe by then, a different aggregation rule would be more preferable?

---

> ### Author Response · Authors · 2025-11-15
> **Response to Reviewer 3rfK**
>
> Thank you for your review of our work. We are very pleased you found our paper to be “extremely well written”, “well motivated”, and that our “analyses tell a cohesive and convincing story about how compute can serve as a reward signal.”
>
> > My biggest complaint is that CaT, as constructed, seems a bit arbitrary \[...\] this isn't a problem if the overarching paper story is more towards whether compute CAN serve as rewards
>
> Our aim is indeed to show compute can serve as rewards. We have revised the introduction to reflect this. We also note that our future work section states “synthesis is just one way of estimating a reference answer; CaT-RL opens the door to reference-free training with task-specific reference estimation strategies”.
>
> > There isn't direct evidence provided by the authors that not including the question in the synthesis prompt actually does anything
>
> Not including the question is not intended to be an important design decision and we have adjusted the introduction to remove the note about keeping the anchor “question-blind”. This note was our original reasoning for not including the question, but it does not make a significant difference to performance. Including the question improves Gemma 3 4B on MATH-500 by just +0.004, well within the margin of error.
>
> > For instance, limiting to yes/no questions might be less expressive than likert scale questions or similar finer-grained ratings
>
> We followed the HealthBench evaluation rubric style which is to ask yes/no questions. Recent work on rubrics as rewards \[Gunjal et al. 2025\] has shown that likert scales for LLM-as-judge do not work as well. However, your point about breaking down individual criteria into likert scales or fine-grained ratings is something that has not yet been explored and could be helpful. The rubric-based reward format can be changed based on task/format if different kinds of scoring end up being better.
>
> > one could imagine doing the rollouts with different prompt rephrasings or conditioning on other differing information
>
> We think it would be interesting to encourage diversity through conditioning on past rollouts for the question, for example. This is something we may explore in follow up work.
>
> > maybe asking the model to filter for diverse responses will yield better combinations?
>
> Yes, as the group size G grows and context lengths become a limit, filtering for better/diverse responses will be important. We see a few potential options for future work such as sampling of a ‘best’ sub-group g from G via rule-based or LLM-based heuristics.
>
> > \[...\] the research question of whether you can use compute as training in general. Therefore I would personally favor that framing.
>
> We are generally in agreement. We have adjusted our motivation in the introduction to make this clearer. Please see the changes in orange in the revised PDF.
>
> > it can no longer improve over the individual rollouts if they tend to agree too much." Do the authors have any ideas on how to potentially alleviate this?
>
> One direction is to use exploration rewards \[Song et al. 2025\] to have an explicit objective that helps maintain diversity throughout training. Another is to self-condition on past rollouts and encourage the model to produce answers that are different. This is a common strategy used in creativity benchmarks \[A\] as well as evolutionary LLM sampling \[B, C\]. We expect these strategies to be helpful in extending the useful duration for which a method like CaT-RL can improve a model reference-free.
>
> > I don't really buy that CaT can meaningfully scale with rollouts beyond a certain extent; \[...\] do the authors believe there will be settings where there are meaningful increases after a reasonably large magnitude
>
> In principle, having more rollouts should only add (not subtract) information and hence always improve. However, like with all test-time scaling, we expect CaT improvements to plateau with larger group size G. As you rightly point out, perhaps more selective methods for aggregation would be needed after a point to continue significant improvements (e.g., the sub-group sampling we proposed in an earlier response).
>
> Thank you again for your thoughtful review. Please let us know if you have any further questions or concerns.
>
> \[A\] McLaughlin, A., Uppuluri, A. and Campbell, J., AidanBench: Evaluating Novel Idea Generation on Open-Ended Questions. In Language Gamification-NeurIPS 2024 Workshop.
>
> \[B\] Romera-Paredes, B., Barekatain, M., et al., 2024\. Mathematical discoveries from program search with large language models. Nature, 625(7995), pp.468-475.
>
> \[C\] Novikov, A., Vũ, N., Eisenberger, M., et al., 2025\. AlphaEvolve: A coding agent for scientific and algorithmic discovery. arXiv preprint arXiv:2506.13131.

---

> > ### Comment · Reviewer_3rfK · 2025-11-20
> >
> > I appreciate the response from the authors, as well as the changes to the introduction. For the first point, may I ask the authors to consider changing the framing throughout the paper (especially in the discussion/conclusion) as well?
> >
> > Otherwise, I consider my concerns addressed. While some additional experiments to satisfy my curiosity would be appreciated, I recognize that the rebuttal time window is short and continue to support the paper.

---

> > > ### Author Response · Authors · 2025-11-22
> > > **Thank you Reviewer 3rfK**
> > >
> > > We appreciate your suggestion and have made additional changes to the discussion/conclusion, results, and method section in light of your suggested framing. We hope this further clarifies the point. Thank you for your continued support of our paper.

---

### Author Response · Authors · 2025-11-16
**Summary of changes**

We are grateful to all the reviewers for their thoughtful comments, suggestions, and positive feedback. Highlights include that our paper is "extremely well-written", "well-motivated within the literature", and our "analyses tell a cohesive and convincing story" (reviewer 3rfK). Our work is "conceptually new", has "experimental breadth and depth", and shows "consistent and notable gains" (reviewer wyte). Additionally, our motivation is "strong" and our synthesis method is "compelling" (reviewer mnEz). Lastly, this work addresses "three important issues" and attempts to tackle "all of these problems simultaneously", synthesis is a "novel contribution" by going beyond self-distillation through novel aggregations, and the "writing and overall presentation is very clear" (reviewer ySqQ).

Below, we summarise the main changes to the paper that we have made in response to reviewer comments. Most modifications have been marked in orange in the updated manuscript.

**New experiments**

- Added **new experiment to ablate rubric generation** to Table 1 (reviewer ySqQ)
- Added **new experiments across three datasets, two models, and up to 10 seeds** to Table 2 and discussion in Result 7 showing that CaT is usually correct to disagree with majority voting and Self-BoN (reviewer mnEz and ySqQ)
- Added Appendix A.1 with **new results on more datasets across 10 seeds for HMMT and AIME'2025** (reviewer mnEz and ySqQ)
- Added Appendix A.2 with **new CaT-RL results against RL with selection method baselines** (reviewer ySqQ)
- Added Appendix C with **new quantitative and qualitative analysis of CaT output traces** against policy rollouts (reviewer ySqQ)

**New discussions/clarifications**

- Emphasised that **CaT-RL delivers better performance than CaT (or selection methods) with 9x less test-time compute**
- Revised the introduction to add **definitions and examples** for readers unfamiliar with RL post-training (reviewer wyte)
- Made it clearer throughout that **CaT-RL is a framework for learning from inference compute** and not only CaT (reviewer 3rfK)
- Added the results of **statistical tests** to Figure 4 and Figure 6 to show that **CaT-RL outperforms CaT and CaT outperforms self-BoN**
- Included details of statistical testing procedures in Appendix H (reviewer mnEz)
- Updated discussion about Figure 6 to highlight that weaker/older models like Llama 3.1 8B may not leverage rollouts as successfully as Gemma/Qwen to beat majority vote (reviewer mnEz)
- Added Appendix A with tabular formatting of all main results (reviewer mnEz)

We hope to continue the discussion further if you have any more points to raise!

---

### Meta-Review · Area_Chair_e7p7 · 2025-12-15

**Summary:**

This paper is with deep discussion between authors and reviewers. So this information helps AC to make a decision a lot. While most of concerns have been addressed, others are still there. Unresolved concerns include: lack of clear intuition, no consistent performance performance over baselines, and problematic experimental setup.

**Reviewer Concerns:**

Reviewer concerns AC thinks were addressed by the rebuttal:

Reviewer 3rfK expressed that his/her concerns have been addressed by the rebuttal.

Reviewer concerns AC believes are still outstanding:

1. Lack of clear intuition behind the “synthesis".

AC's comment: Reviewer wyte is not satisfied with the rebuttal on the intuition of the paper.

2. From both the initial results from the paper and the additional experiments collected during the rebuttal phase, the framework does not seem to offer a consistent performance improvement over simpler baselines.

AC's comment: This is Reviewer mnEz's comment after the rebuttal.

3. Reviewer ySqQ noticed that the authors' results for HMMT and AIME have (1) scores that seem incorrectly low for Qwen-3 4B*, even when using non-thinking mode, and (2) are probably benchmarked on too few seeds, as the datasets are very small.

AC's comment: The reviewer still has concern on the experimental setup.

**Reviewer Scores:**

Reviewer wyte will increase the score, while AC thinks other reviewers may keep their scores. With high probability, AC thinks no reviewer will champion the paper, while there is still negative scores. Given that, AC would recommend rejection of this paper.

---

### Decision · Program_Chairs · 2026-01-26

Reject